# Towards Protection of Nucleic Acids from Herbicide Attack: Self-Assembly of Betaines Based on Pillar[5]arene with Glyphosate and DNA

**DOI:** 10.3390/ijms24098357

**Published:** 2023-05-06

**Authors:** Anastasia Nazarova, Pavel Padnya, Arthur Khannanov, Aleksandra Khabibrakhmanova, Pavel Zelenikhin, Ivan Stoikov

**Affiliations:** 1A. M. Butlerov Chemistry Institute, Kazan Federal University, 18 Kremlyovskaya Str., 420008 Kazan, Russia; padnya.ksu@gmail.com (P.P.);; 2Institute of Fundamental Medicine and Biology, Kazan Federal University, 18 Kremlyovskaya Str., 420008 Kazan, Russia; sasha_xb@mail.ru (A.K.); pasha_mic@mail.ru (P.Z.); 3Federal Center for Toxicological, Radiation, and Biological Safety, Nauchny Gorodok-2, 420075 Kazan, Russia

**Keywords:** pillar[5]arene, glyphosate, amino acid, DNA protection, association

## Abstract

Herbicides are one of the main parts of pesticides used today. Due to the high efficiency and widespread use of glyphosate-based herbicides, the search for substances reducing their genotoxicity is an important interdisciplinary task. One possible approach for solving the problem of herbicide toxicity is to use compounds that can protect DNA from damage by glyphosate derivatives. For the first time, a method for developing DNA-protecting measures against glyphosate isopropylamine salt (GIS) damage was presented and realized, based on low-toxicity water-soluble pillar[5]arene derivatives. Two- and three-component systems based on pillar[5]arene derivatives, GIS, and model DNA from salmon sperm, as well as their cytotoxicity, were studied. The synthesized pillar[5]arene derivatives do not interact with GIS, while GIS is able to bind DNA from salmon sperm with lgKa = 4.92. The pillar[5]arene betaine derivative containing fragments of L-phenylalanine and the ester derivative with diglycine fragments bind DNA with lgKa = 5.24 and lgKa = 4.88, respectively. The study of the associates (pillar[5]arene-DNA) with GIS showed that the interaction of GIS with DNA is inhibited only by the betaine pillar[5]arene containing fragments of L-Phe (lgKa = 3.60). This study has shown a possible application of betaine pillar[5]arene derivatives for nucleic acid protection according to its competitive binding with biomacromolecules.

## 1. Introduction

The global population will grow almost by 2 billion by 2050, according to the United Nations report [1]. This increase in population will lead to the growth of total agricultural production by 60% compared to 2005. However, the situation with food security deteriorates every year due to a reduction of plowed field areas and environmental problems (e.g., soil erosion, climate change, water availability) [2,3,4]. Existing modern growth promoters and pesticides can only partially solve the problem of the food crisis due to their use limits (high toxicity and non-directional action). 

Most of the pesticides used in agriculture do not reach crops. This is due to a dispersion of pesticides into the environment caused by their degradation, photolysis and volatilization [5,6]. Herbicides are one of the main parts (almost 40%) of pesticides used today [7]. Herbicides quickly enter water sources and soil causing serious environmental damage [8,9]. Glyphosate (*N*-(phosphonomethyl)glycine) is one of the widely used active ingredients in commercially known available herbicides [10,11,12]. The WHO recognized glyphosate as a low-toxic compound in 1997, while glyphosate-based commercial formulations are one–two orders of magnitude more toxic [13,14]. Over the past 20 years, it has been established that glyphosate and pesticides based on it (for example, roundup) have an apparent genotoxicity. They are capable of interacting with DNA leading to its damage [15,16,17,18,19]. One possible approach for solving the problem of herbicide toxicity is to use compounds that can protect DNA from damage by glyphosate derivatives. It was shown in 2022 by Alvarez-Moya et al. [20], that the addition of various antioxidants (e.g., ascorbic acid and resveratrol) helps to reduce the damage of DNA from *Oreochromis niloticus*, *Ambystoma mexicanum* erythrocytes, and human lymphocytes. Due to the high efficiency and widespread use of glyphosate-based herbicides, the search for substances reducing their genotoxicity is an important interdisciplinary task of modern science. 

The use of synthetic macrocyclic compounds such as cucurbit[n]urils [21,22] and (thia)calix[n]arenes [23,24] for the binding of various organic pollutants has acquired a strong practical interest due to their properties (low toxicity, ease of preparation, and functionalization, as well as the possibility of scaling up their synthesis). New applications of macrocyclic compounds were demonstrated by pillar[n]arenes discovered by Ogoshi in 2008 [25]. This type of synthetic receptors is characterized by all the qualities mentioned above for (thia)calixarenes and cucurbiturils. Moreover, pillar[n]arenes have a macrocyclic cavity similar to cyclodextrins known for more than 100 years [26,27,28]. Pillar[5]arenes are used as a part of electrochemical sensors [29,30,31,32,33,34,35,36] that promote paraquat and its derivatives detection of the formation of host-guest complexes. However, the number of works dedicated to the use of pillar[n]arenes and their derivatives as antidotes for a series of herbicides is limited [37,38,39], despite the relevance of the research topic. At the same time, the impact of adding macrocyclic compounds to pesticides on their genotoxicity has not been assessed. 

In this work, an approach for the creation of DNA protection systems from the damage by glyphosate isopropylamine salt (GIS) based on low-toxicity water-soluble derivatives of pillar[5]arene was proposed and implemented for the first time. Two- and three-component systems based on pillar[5]arene derivatives, GIS, and model DNA from salmon sperm, as well as their cytotoxicity, were studied. The results obtained open up new possibilities for protecting nucleic acids from damaging factors. 

## 2. Results and Discussion

### 2.1. Synthesis of Decasubstituted Pillar[5]arenes Containing Betaine Fragments

Amino acids and peptides are well-known building blocks for various supramolecular architectures playing a key role in biological systems. These strictly specified architectures, based on peptide derivatives, are important for understanding the processes of biological self-assembly, and still remain one of the urgent topics of supramolecular chemistry [40]. Thus, the spatial structure of protein molecules is of great interest in medicinal chemistry [41], for example, in the treatment of neurodegenerative diseases (Alzheimer’s disease, dementia, etc.). Among synthetic receptors presented to date, the most convenient synthetic “platforms” for the development of biomimetic systems are macrocyclic compounds [42,43,44]. The macrocyclic structure opens up the possibility of optimizing the geometry of binding sites in the receptor to achieve complementary host-guest interaction. Furthermore, a combination of interaction centers of different natures within a macrocyclic system can lead to the creation of unique three-dimensional conformationally mobile structures. 

Molecular recognition in water is a basis for various vital processes. A significant number of synthesized macrocyclic compounds (cyclodextrins, cucurbiturils, calixarenes, and their thia-analogues, pillararenes) are capable of binding guest molecules in water [45,46,47,48,49,50]. Furthermore, the introduction of small peptide fragments into the structure of macrocycles has shown the possibility of creating water-soluble receptors, which adopt a well-defined and precisely controlled secondary structure [51,52]. 

Macrocycles **1** and **2** containing quaternary ammonium, glycylglycine, and *L*-phenylalanine ester fragments [53] were used for obtaining betaine pillar[5]arene derivatives. The choice of the *L*-phenylalanine residue has been made due to the presence of a benzyl group in the amino acid structure, which can enhance the interaction with guest molecules due to π-stacking and hydrophobic effects. In addition, the formation of α-helices due to hydrogen bonds and Van der Waals forces has been shown for phenylalanine, which leads to a denser packing of amino acid fragments [54,55]. Diglycine was chosen because of the presence of several amide moieties that are prone to hydrogen bonding, which can affect host coordination upon binding to substrates. Thus, we assumed that the modification of pillar[5]arene by GlyGly and Phe would lead to the creation of a new type of amphiphiles, as well as the emergence of a synergetic effect due to the combination of macrocyclic platform and ten amino acid residues to achieve an effective interaction with the substrate. 

The reaction of macrocycles **1** and **2** with lithium hydroxide was studied in a THF-water solvent mixture (*v*/*v* = 1:2) for 12 h at room temperature (Figure 1). Target compounds **3** and **4** were obtained with a yield of 94% and 90%, respectively. 

^1^H and ^1^H–^1^H NOESY NMR spectra of compound **4** is shown in Appendix A and 1a. The amide group’s signals appear as two multiplets at 9.37–9.64 (H^10^) and 8.42–8.71 (H^4^) ppm. The proton signals of the phenyl fragments resonate as a multiplet at 7.14–7.30 ppm. The macrocycle aromatic protons (H^1^) signals also appear as a broadened singlet at 6.85 ppm. The oxymethylene (H^3^) and methylene (H^5^) protons signals resonate as a multiplet at 4.31–4.45 ppm. Methylene bridges protons (H^2^) signals also appear as a multiplet in the 3.75–3.87 ppm region. The proton signals of the methyl (H^8^) and methylene (H^7^) groups resonate as a multiplet in the 3.07–3.19 ppm region. The signals of the methylene protons of the benzyl and propylidene fragments appear as multiplets in 2.86–2.91 (H^12^) and 1.85–2.02 (H^6^) ppm regions, respectively. 

The study of the spatial structure of macrocycle **4** by 2D ^1^H-^1^H NOESY NMR spectroscopy (Figure 1a) showed the presence of the cross peaks between H^6^ protons of propylidene fragment and H^12^ methylene protons, as well as H^13^ protons of the benzyl fragment. Additionally, the spectrum contains correlations between H^8^ methyl protons and H^10^ amide protons, as well as H^11^ protons of the methyne group. The cross peaks appear between H^9^ methylene protons at the quaternary nitrogen atom and H^11^ methyne protons. The presence of these cross peaks above is probably due to the spatial proximity of the charged parts in the oligopeptide structure of the substituents in such a way that the α-helix fragment is formed (Figure 1b). The comparison of the one-dimensional ^1^H NMR spectrum of pillar[5]arene **4** with the ^1^H NMR spectrum of ester derivative **2** (Appendix A) additionally confirms the reaction completeness. There are no signals of ethoxy groups of the amino acid substituent in the ^1^H NMR spectrum (Appendix A). The correlations between methyne protons H^k^ and methylene protons H^g^ and H^f^ in the ^1^H-^1^H NOESY NMR spectrum of macrocycle **2** are observed (Appendix A). The cross peaks between methylene H^e^ protons and H^j^ amide protons are additionally observed. It indicates spatial proximity within the amino acid substituent and the formation of an α-helix fragment (Appendix A). 

The IR spectrum of compound **4** also confirms the successful hydrolysis of pillar[5]arene **2**, as indicated by the shift and the intensity of the characteristic band of the carbonyl stretching vibrations of the carboxyl fragment from 1735 cm^–1^ for compound **2** to 1726 cm^–1^ for macrocycle **4** (Figure 1c). Moreover, the shift of this carbonyl band is probably associated with the spatial proximity of the charged parts in the oligopeptide structure of the substituents in such a way that a six-membered cycle is formed. The amide I band shifted by 15 cm^–1^ for betaine derivative **4** (1660 cm^–1^). Furthermore, it broadened and increased in intensity, which indicates the associate’s formation. In addition, there is an overlap of the amide II and the carboxylate anion (1550 cm^–1^) band for macrocycle **4**, while the amide II of ester derivative **2** is located at 1537 cm^–1^. 

A different spatial packing of the molecule is observed for diglycine derivative **3** (Appendix A). Thus, the cross peaks between H^1^ protons of the aromatic fragments and H^8^ methylene protons at the quaternary nitrogen atom, as well as H^11^ protons (Appendix A), are observed in the ^1^H–^1^H NOESY NMR spectrum of compound **3**. It is also worth noting the presence of the correlations between H^5^ methylene protons and H^12^ amide protons, as well as between H^6^ methylene protons of the propyl fragment and H^13^ protons. The presence of the cross peaks above in the ^1^H–^1^H NOESY NMR spectrum of compound **3** suggests that the substituents on both rims of the macrocycle are spatially close to each other (Appendix A). Such proximity on both rims of the pillar[5]arene is probably due to electrostatic interaction between the positively charged quaternary nitrogen atom and the carboxyl fragment, and the formation of hydrogen bonds between the two substituents. 

The study of the IR spectra of compounds **1** and **3** showed a shift of the amide I band from 1656 cm^–1^ to 1630 cm^–1^, respectively, as well as its broadening in the case of pillar[5]arene **3**, which indicates the formation of intramolecular hydrogen bonds (Figure 2). The amide II band shifted from 1538 cm^–1^ for ester derivative **1** to 1554 cm^–1^ for macrocycle **3**. Previously, the dependence of the amide I band shift in the IR spectrum of a compound on the secondary structure of a protein (α-helix, β-sheets, or β-turn) was shown in a group of Ivan Usynin [56]. It was exhibited that the amide I band for α-helices is in 1660–1649 cm^–1^ region, while for β-sheets, the amide I band shifts to 1637–1614 cm^–1^ region. The amide I band of macrocycles **1**, **2**, and **4** is in the region characteristic of α-helices (1656, 1675, and 1660 cm^–1^, respectively). The amide I band appears at 1630 cm^–1^ for pillar[5]arene **3**, which indicates the formation of β-sheets by the oligopeptide substituents of compound **3**. The described secondary structure of oligopeptide substituents in macrocycles **1**–**4** is confirmed by the two-dimensional ^1^H–^1^H NOESY NMR spectroscopy data (Figure 1 and Appendix A).

Thus, the betaine derivatives based on pillar[5]areness were synthesized in good yields. The structure of the synthesized compounds has been fully characterized (Appendix A) by a series of physical methods (^1^H, ^13^C NMR and IR spectroscopy, and mass spectrometry). Based on the findings revealed by 2D ^1^H–^1^H NOESY NMR and IR spectroscopy, the obtained macrocycles exhibit properties more characteristic of peptides and oligopeptides rather than of the expected dendrimer-like spatial structure. The macrocycle containing diglycine residues is prone to the spatial proximity of positively and negatively charged substituents fragments of both macrocycle rims, as well as their proximity to hydroquinone fragments of the macrocycle, which leads to the formation of a β-sheet fragment, whereas the pillar[5]arene containing *L*–Phe residues is characterized by a formation of an α-helix fragment inside the substituent due to the spatial convergence of the positively charged nitrogen atom and the carboxyl fragment. 

### 2.2. Aggregation Properties of the Synthesized Macrocycles

The next stage of the work was the study of aggregation properties of both previously synthesized ester derivatives of amino acids **1** and **2**, and macrocyclic betaine derivatives **3** and **4** by dynamic light scattering (DLS). The study was carried out in water in the 1 × 10^–6^–1 × 10^–4^ M concentration range (Appendix A). The size of associates increases with decreasing macrocycle **1** concentration in solution (Table 1). A different pattern is observed for betaine derivative **3**. Thus, the particles become coarser, and the polydispersity index (PDI) of the systems increased with an increase in compound **3** concentration, which is probably due to the formation of β-sheets by oligopeptide substituents. The presence of two oppositely charged fragments in one substituent leads to the fact that compound **3** becomes similar to surfactant molecules in terms of its aggregation properties. 

Compound **4** is also characterized by the formation of polydisperse systems, regardless of concentration. The stable associates are formed in the total concentrations range only in the case of macrocycle **2** containing ester fragments of phenylalanine. An insignificant coarsening of particles from 144 nm (1 × 10^–6^ M) to 179 nm (1 × 10^–4^ M) is observed with an increase of pillararene **2** concentration. The formation of the stable aggregates of compound **2** is explained by the presence of lipophilic benzyl fragments in the substituents, which stabilizes the formed particles. The hydrolysis of macrocycle **2** leads to the appearance of the charges in the substituent of pillar[5]arene **4**, which are spatially close to each other (Figure 1b). In this view, there are no stable associates in the solution. 

The nanosized aggregates formed by macrocycle **2** were studied by transmission electron microscopy (TEM). The formation of the spherical nanosized particles with sizes close to those determined by DLS was confirmed. Figure 3 shows the formation of particles with 50 nm average diameter, which stick together and form larger aggregates.

### 2.3. Complexation of the Pillar[5]arenes Containing Peptide and Betaine Fragments with DNA and Glyphosate Salt

Herbicides are widely used in agriculture to control weeds despite their negative effect on the environment [37]. However, it is known that herbicides can cause acute and chronic poisoning when exposed to organisms. Glyphosate is a non-selective systemic herbicide (Figure 4), ranked first in terms of production [57]. Therefore, roundup (glyphosate-based formulation) has become widespread in agriculture. The main active ingredient of roundup is an isopropylamine salt of glyphosate (Glyphosate isopropylamine salt, GIS) (Figure 4).

The interaction of GIS with a series of synthesized pillar[5]arenes **1**–**4** and DNA was studied by UV-Vis spectroscopy. UV-Vis spectroscopy is a universal method to study the complex properties of synthetic receptors. The interaction of GIS with DNA from salmon sperm was initially studied (Figure 5). DNA from salmon sperm was chosen as a model substrate due to its relatively low molecular weight and small size. The experiments were carried out in a Tris-HCl buffer system (pH = 7.44).

It was found that the herbicide addition to the DNA solution leads to a hypochromic effect at 260 nm. As a result, the intensity of the DNA absorption band decreased (Figure 5a). The decrease in the absorption intensity (hypochromic effect) of DNA upon interaction with the analyte can be explained by the compaction of the biomacromolecule, according to the data [58]. 

The association constants were determined by spectrophotometric titration to quantify the interaction of DNA with GIS. The UV-Vis spectra of the DNA–GIS system were recorded in such a way that the DNA concentration remained constant (2.5 × 10^–7^ M), and the herbicide concentration ratio increased from 1:0.3 to 1:10 (Figure 5b). The obtained experimental data were processed by the BindFit program. The association constant of a 1:1 complex was calculated (Appendix A). The logarithm of the association constant of DNA with GIS was determined as lgK_a_ = 4.92. The stoichiometry of the complex was also confirmed by titration data processed using host:guest ratios of 1:2 and 2:1. However, in this case, the association constant of the complexes was determined with a large error. 

The next stage was the study of the complexation of synthesized pillar[5]arenes **1**–**4** with GIS by UV-Vis spectroscopy. However, no changes in the UV-Vis spectra were shown for all the studied macrocycles (Appendix A). Therefore, synthesized pillar[5]arenes **1**–**4** did not interact with GIS. 

Further, the interaction of macrocycles **1**–**4** with the DNA from salmon sperm in Tris-HCl was studied by UV-Vis spectroscopy. A hypochromic effect was observed when pillar[5]arenes **1** and **4** solutions were added to the DNA solution (Figure 6a and Appendix A). The association constant of compound **4**-DNA complex was determined by spectrophotometric titration. The DNA concentration remained constant (C_DNA_ = 2.5 × 10^–7^ M), and the macrocycle concentration increased from the ratio of 1:1 to 1:10 (Figure 6b). The linearization of the resulting titration curves by the BindFit program allowed us to calculate the association constants of the 1:1 complex (Appendix A). The logarithm of the association constant of pillar[5]arene **4** with the DNA was determined as lgK_a_ (**4**–DNA) = 5.24. 

The association constant of compound **1**-DNA complex was also determined by spectrophotometric titration (Appendix A). The logarithm of the association constant of pillar[5]arene **1** with DNA was determined as lgK_a_(1–DNA) = 4.88 (Appendix A). A small difference in the logarithms of the association constants of DNA with macrocycles **1** and **4** indicates that the nature of the amino acid substituent does not influence the binding of the biomacromolecule and is determined by the charge of the macrocycle. 

It is well known that water-soluble derivatives of pillar[5]arene can applicate as antidotes for various drugs [59,60]. The interaction of GIS with DNA with a logarithm of the association constant of 4.92 was shown. It was also found that pillar[5]arenes **1** and **4** are able to bind DNA with lgK_a_ 4.88 and 5.24, respectively. In this regard, a hypothesis about the possibility of using the synthesized macrocycles as a glyphosate antidote for DNA protection was proposed. Therefore, the behavior of the associate (macrocycle-DNA) with GIS was studied by UV-Vis spectroscopy to test the hypothesis above. First, a solution of DNA (2.5 × 10^–7^ M) was added to the macrocycle (**1**–**4**) solution (3 × 10^–6^ M). Then the herbicide solution (C_GIS_ = 3 × 10^–6^ M) was added to the resulting associate, and the absorption spectrum of the resulting ternary system was recorded. The absorption spectrum of this three-component system differed from the additive spectrum (Figure 7a,b and Appendix A), and a hyperchromic effect was observed for all three-component systems. This effect is due to the competitive interaction of DNA with the glyphosate salt, as a result of which the biomacromolecule is decompacted.

Further, the interaction of the (pillar[5]arene–DNA) associated with GIS was quantified (Figure 7c,d and Appendix A). The concentration of the (macrocycle–DNA) associate remained constant (C_PA_ = 3 × 10^–6^ M, C_DNA_ = 2.5 × 10^–7^ M), while the herbicide concentration increased from the ratio of 1:0.3 to 1:10 relative to the concentration of the associate in the course of spectrophotometric titration for all systems. The obtained experimental data were processed by the BindFit program (Appendix A). The effective association constants of the (macrocycle-DNA)-GIS associates of 1:1 composition were calculated (Table 2).

It was found that the logarithm of the association constant for the (pillar[5]arene-DNA)-GIS associates in the case of compounds **1**–**3** slightly differ from the logarithm of the association constant of DNA-GIS (4.92), while for the associate formed by macrocycle **4** with DNA, a decrease in lgK_a_ by more than an order of magnitude was shown, which indicates the inhibition of the interaction of GIS with DNA. Apparently, macrocycle **4** competes with GIS for the interaction with DNA (lgK_a_(**4**-DNA) = 5.24; lgK_a_(DNA-GIS) = 4.92).

The obtained (macrocycle-DNA)-GIS associates were also studied by DLS in Tris-HCl buffer solution. The concentration of each component of the mixture was the same as in the experiments performed by UV-Vis spectroscopy (C_PA_ = 3 × 10^–6^ M, C_DNA_ = 2.5 × 10^–7^ M, C_GIS_ = 3 × 10^–6^ M). The formation of the stable aggregates (d = 183 ± 5 nm, PDI = 0.19, and ζ = –22.2 mV) was found only for the system based on macrocycle **2** with phenylalanine ester fragments (Appendix A). It was previously shown that pillar[5]arene **2** forms the most stable aggregates (Table 1) due to the secondary structure of the substituents (α-helix) and the presence of uncharged terminal ester fragments. This, apparently, leads to the formation of stable aggregates in the case of the three-component (macrocycle-DNA)-GIS systems. The associate’s formation for three-component (**2**–DNA)–GIS system was additionally confirmed by TEM (Figure 8). It was shown that spherical nanosized particles with sizes close to those determined by DLS are formed.

### 2.4. Cytotoxicity and Apoptosis-Inducing Ability of the Synthesized Macrocycles in the Absence and Presence of GIS

The next stage of this work was the study of pillar[5]arenes **1**–**4** ability to inhibit the viability and proliferative activity of embryonic lung epithelial (*LEC*) and human lung adenocarcinoma (*A549*) cells using the MTT test [61]. It was found that macrocycles **1**, **3**, and **4** did not show statistically significant cytotoxic activity against *LEC* and *A549* cells (Appendix A) in the studied concentration range (2–100 μg/mL). It was also shown that pillar[5]arene **2** reduced the viability of *A549* and *LEC* cells at the concentration of ≥5 μg/mL (Figure 9a,b).

The average inhibitory concentration (IC_50_) was calculated for compounds **1**–**4**. The data are presented in Table 3. The cytotoxic activity of the macrocyclic compounds against *LEC* cells is consistent with those obtained for *A549* cells.

The cytotoxic effect of pillar[5]arene **2** containing ester fragments of phenylalanine is probably due to the stable aggregates’ formation (Table 1). Pillar[5]arene **2** was tested as a potential apoptogen, as it showed the ability to reduce the viability of *A549* and *LEC* cells [62]. It has been suggested that the decrease in viability identified in the MTT assay is mediated by the induction of programmed cell death by this macrocyclic compound. The agent concentration of 25 μg/mL was chosen for the experiment. The results of the cytometric analysis are shown in Figure 9f. It was found that pillar[5]arene **2** induced apoptosis in both *A549* and *LEC* cells (Figure 9f). The apoptosis-inducing effect of compound **2** on *A549* and *LEC* cells turned out approximately the same. The proportion of living cells in the population was 74.8% and 72.7%, respectively, while the value of this indicator for the variant without treatment was significantly higher and amounted to 93.1% and 98.3%, respectively. Thus, it was shown that synthesized pillar[5]arenes **1** and **3**, **4** did not exhibit statistically significant cytotoxic activity toward *LEC* and *A549* cells in the concentrations range of 2–100 μg/mL. The IC_50_ of macrocycle **2** containing phenylalanine ester fragments was found at 29.67 and 28.16 µg/mL concentration for *A549* and *LEC* cells, respectively. 

Further, the ability of GIS to inhibit the viability and proliferative activity of *LEC* and *A549* cells was studied by the MTT test [61]. It was shown that GIS did not have the ability to reduce the viability of *LEC* and *A549* cells at a concentration of ≤5 mg/mL. The cytotoxic effect of GIS on *LEC* and *A549* cells was recorded only at a concentration of ≥10 mg/mL (Figure 9c,d). The average inhibitory concentration (IC_50_) was calculated for GIS (Table 3). The IC_50_ of GIS values were 11.43 mg/mL and 10.18 mg/mL, respectively. 

The ability to inhibit the viability and proliferative activity of *A549* cells of macrocycles **1**–**4** in the presence of GIS was studied using the MTT assay. It was found that the combined action of studied pillararenes **1**–**4** (1 × 10^–5^ M) with GIS (3.47 × 10^–2^ M) resulted in a slight decrease in toxicity in the case of compound **1** (Figure 9e), which tends to form complexes with DNA. The developed approach opens up the possibility of creating pillar[5]arene-based herbicide antidotes. At the same time, an increase in toxicity was shown for the macrocycle **2**–GIS system, which is due to the toxicity of pillararene itself. No significant differences were found for compounds **3** and **4**. 

## 3. Materials and Methods

### 3.1. General Experimental Information

Detailed information on the equipment, methods, and physical–chemical characterization is presented in the Appendix A.

### 3.2. General Procedure for the Synthesis of Compounds ***3*** and ***4***

In a round-bottom flask equipped with a magnetic stirrer, 0.15 g (0.029 mmol for **1** and 0.031 mmol for **2**) of pillar[5]arene **1** (**2**) was dissolved in 6 mL of THF-H_2_O mixture (V_THF_/V_H2O_ = ½). Then, 2.9 (3.1) mmol of lithium hydroxide was added. The reaction mixture was mixed for 12 h at room temperature (25 °C). The precipitates were filtered. The solvent was removed under reduced pressure. The products were recrystallized from 2–propanone. The precipitates were dried under reduced pressure over P_2_O_5_.

#### 3.2.1. 4,8,14,18,23,26,28,31,32,35-Decakis-[(*N*-[3’-(dimethyl{[(oxidocarbonylmethyl)aminocarbonylmethyl]aminocarbonylmethyl}ammonio)propyl]aminocarbonylmethoxy]-pillar[5]arene (**3**)

Yield 0.109 g (94%)—yellow viscous oil.

^1^H NMR (400 MHz, 298 K, DMSO-*d_6_*): δ 1.93–2.05 (m, 20H, NHCH_2_CH_2_CH_2_N^+^), 3.26 (br.s, 60H, N^+^(CH_3_)_2_), 3.68–3.77 (m, 40H, NHCH_2_C(O) and NHCH_2_C(O)O^−^), 3.78–3.87 (m, 40H, NHCH_2_CH_2_CH_2_N^+^ and N^+^CH_2_C(O)), 4.24–4.36 (m, 20H, NHCH_2_CH_2_CH_2_N^+^), 4.40–4.52 (m, 20H, ArOCH_2_), 6.83 (br.s, 10H, ArH), 8.44–8.60 (m, 20H, OCH_2_C(O)NH and C(O)NH), 9.37–9.55 (m, 10H, NHCH_2_C(O)O^−^). 

^13^C NMR (100 MHz, 298 K, DMSO-*d_6_*): δ 20.6, 20.7, 22.9, 31.3, 35.9, 36.4, 36.5, 41.1, 41.5, 42.2, 50.5, 51.6, 60.3, 60.6, 61.2, 61.4, 62.3, 62.5, 63.4, 64.1, 67.9, 114.9, 128.3, 149.3, 163.0, 163.8, 164.0, 166.4, 168.7, 169.0,169.3, 171.1, 171.4, 171.6. 

IR, ν/cm^–1^: 1554(N-H), 1630 (C=O), 1734 (C=O), 3359 (NH). 

HRMS (ESI), *m/z*: Calculated for: 626.4702 [M+6 H^+^]^6+^, 715.5628 [M+5 H^+^]^5+^, 939.2016 [M+4 H^+^]^4+^; found: 626.4720 [M+6 H^+^]^6+^, 715.5618 [M+5 H^+^]^5+^, 939.2016 [M+4 H^+^]^4+^.

#### 3.2.2. 4,8,14,18,23,26,28,31,32,35-Decakis-[(*N*-[3’-dimethyl({oxidocarbonyl[*S*-benzyl]methyl}aminocarbonylmethyl)ammonio]propyl)aminocarbonylmethoxy]-pillar[5]arene (**4**)

Yield 0.107 g (90%)—yellow viscous oil.

^1^H NMR (400 MHz, 298 K, DMSO-*d_6_*): δ 1.83–2.02 (m, 20H, NHCH_2_CH_2_CH_2_N^+^), 2.81–2.90 (m, 20H, CH_2_Ph), 2.98–3.19 (m, 80H, NHCH_2_CH_2_CH_2_N^+^ and N^+^(CH_3_)_2_), 3.80 (br.s, 10H, ArCH_2_Ar), 4.01–4.17 (m, 10H, CHCH_2_Ph), 4.19–4.29 (m, 20H, N^+^CH_2_C(O)), 4.29–4.53 (m, 40H, ArOCH_2_ and NHCH_2_CH_2_CH_2_N^+^), 6.85 (br.s, 10H, ArH), 7.16–7.30 (m, 50H, Ar^Ph^H), 8.40–8.58 (m, 10H, NHCH_2_CH_2_CH_2_N^+^), 9.37–9.54 (m, 10H, C(O)NHCH). 

^13^C NMR (100 MHz, 298 K, DMSO-*d_6_*): δ 22.7, 30.9, 35.7, 36.5, 51.2, 54.7, 62.0, 63.2, 67.6, 114.7, 126.5, 128.3, 129.4, 137.9, 149.1, 162.6, 163.1, 168.7, 172.6. 

IR, ν/cm^–1^: 1550 (N-H), 1660 (C=O), 1726 (C=O), 2963 (OH), 3064 (OH), 3364 (NH). 

HRMS (ESI), *m/z*: Calculated for: 681.6799 [M+6 H^+^]^6+^, 817.8144 [M+5 H^+^]^5+^, 1022.0162 [M+4 H^+^]^4+^; found: 681.6815 [M+6 H^+^]^6+^, 817.8153 [M+5 H^+^]^5+^, 1022.0159 [M+4 H^+^]^4+^.

## 4. Conclusions

In the course of this study, water-soluble betaine derivatives of pillar[5]arene containing amino acid residues (GlyGly and *L*-Phe) were synthesized for the first time. It was shown that the obtained macrocycles exhibit properties more characteristic of peptides and oligopeptides instead of the expected dendrimer-like spatial structure. Thus, decasubstituted pillar[5]arene **3** containing diglycine residues is prone to the spatial proximity of positively and negatively charged substituents of both rims, as well as their proximity to hydroquinone fragments of the macrocycle, which leads to the formation of a β-sheets. Macrocycle **4** containing *L*-Phe residues is characterized by the formation of an α-helix inside the substituent due to the spatial proximity of the positively charged nitrogen atom and the carboxyl fragment. It was found that macrocycles **1**, **3**, and **4** did not show statistically significant cytotoxic activity to *LEC* and *A549* cells in the concentrations range of 2-100 μg/mL. The IC_50_ of macrocycle **2** was found at 29.67 and 28.16 μg/mL concentration for *A549* and *LEC* cells, respectively. This occurs due to the formation of stable aggregates by macrocycle **2** because of the presence of the uncharged terminal fragments and the formation of an α-helix by the oligopeptide substituent. 

It was shown that the synthesized pillar[5]arene derivatives did not interact with glyphosate salt (GIS), while GIS was able to bind DNA from salmon sperm with lgK_a_ = 4.92. It was also found that betaine derivative of pillar[5]arene **4** containing fragments of *L*-phenylalanine bound DNA with lgK_a_ = 5.24, and ester derivative **1** with diglycine fragments bound DNA with lgK_a_ = 4.88. It was found that associate (pillar[5]arene **4**–DNA) inhibited (lgK_a_ = 3.60) the interaction of GIS with DNA, and that the combined action of the studied pillararenes with glyphosate resulted in a slight decrease in toxicity in the case of compound **1** which tends to form complexes with DNA. The developed approach opens up the possibility of creating pillar[5]arene-based herbicide antidotes. Thus, the competition between the macrocycle and glyphosate for the interaction with DNA will potentially protect DNA molecules from the herbicides’ effects.

## Data Availability

The data presented in this study are available in the Appendix A.

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
