# Peer review of "Towards Protection of Nucleic Acids from Herbicide Attack: Self-Assembly of Betaines Based on Pillar[5]arene with Glyphosate and DNA"

_ijms, 2023, doi:10.3390/ijms24098357_

Round 1
Reviewer 1 Report
In this paper, a novel approach is presented to use of pillar[n]arenes and their derivatives as antidotes for herbicides. Authors has studied the cytotoxicity by adding macrocyclic compounds to pesticides. This study has shown an application of betaine pillar [5] arene derivatives for nucleic acids protection according to its competitive binding with biomacromolecules, and open new possibilities for protecting nucleic acids from damaging factors.
The paper is professionally written, and the conclusions are supported by good experimental results. The paper addresses relevant environmental concerns and provide information which can be helpful for the readers, and therefore, should be publishes. I recommend accepting is paper for publication.
English language is fine, minor editing is required.
Author Response
English language had been checked by native speaker, necessary changes have been made.
Reviewer 2 Report
Excellent work, I recommend improving the sharpness of figure 1
I have no suggestions in this section
Reviewer 3 Report
In this article entitled “Towards protection of nucleic acids from herbicide attack: self-assembly of betaines based on pillar[5]arene with glyphosate and DNA”, the authors synthesized several derivatives of pillar[5]arene and studied their effect on protecting DNA from the damage by glyphosate isopropylamine salt (GIS). The structures of the synthesized molecules were characterized and tested for binding with DNA. However, the current data are not sufficient to support some of the conclusions in the manuscript which need to be addressed:
11. In Figure 3, the TEM image looks more likely to be crystalline instead of assemblies. And the size (50 nm) is quite different from DLS results (144-179 nm), which is opposite to what the authors claimed.
22. The authors studied the assemblies of pillararenes. Why do the authors think it is important? Are the inhibition results based on monomers or aggregates?
33. Considering that the significant difference between concentrations of pillararenes 1–4 (1 × 10–5 M) and GIS (3.47 × 10–2M) used for the cell viability tests and their binding affinity with DNAs, the pillararenes seems insufficient to inhibit the binding between GIS and DNAs. Thus, the reviewer doesn’t think the current cell experiment results support the conclusions.
44. In table 3, the IC50 of GIS is much higher than pillararenes. Why do the authors use more toxic compounds to inhibit another binding and avoid DNA damage if they have similar binding affinity to the DNA?
55. In Figure 7, how about the control experiment data from DNA only?
Author Response
First of all, we would like to thank respected Reviewer for careful consideration of the manuscript. In accordance with the comments, the following changes have been made:
- In Figure 3, the TEM image looks more likely to be crystalline instead of assemblies. And the size (50 nm) is quite different from DLS results (144-179 nm), which is opposite to what the authors claimed.
Zoom in of the TEM images allows to see that it is exactly referred to associates, not to crystal structures. Since the associates do not have clear boundaries and strictly defined forms. The difference in the sizes of associates determined by TEM and DLS is primarily due to a presence of a solvent coat in the case of DLS, which is “burned out” while performing TEM experiments.
TEM images of aggregates formed by pillararene 2 (1 × 10-4 M) in water.
- The authors studied the assemblies of pillararenes. Why do the authors think it is important? Are the inhibition results based on monomers or aggregates?
We are grateful the referee for the questions. It is well known that aggregation of drugs significantly affects their activity (10.3390/molecules25122850; 10.1016/j.ejmech.2008.09.037; 10.1016/j.msec.2020.110930; 10.1021/acs.chemrev.6b00562; 10.1021/cb300189b). Obtaining associates based on peptide derivatives due to non-covalent interactions are important for understanding the processes of biological self-assembly and still remain one of the urgent topics of supramolecular chemistry. Conducted experiments indicate that an inhibition of interaction DNA with glyphosate is indeed due to the formation of the pillar[5]arene-DNA association. This is clearly proven by the data obtained by UV spectroscopy for macrocycle 1 (lgKa(1–DNA) = 4.88). In addition, toxicity of macrocycle 2 to A549 and LEC cells (29.67 and 28.16 μg/mL, respectively) is due to the formation of stable aggregates by compound 2.
- Considering that the significant difference between concentrations of pillararenes 1–4 (1 × 10–5 M) and GIS (3.47 × 10–2M) used for the cell viability tests and their binding affinity with DNAs, the pillararenes seems insufficient to inhibit the binding between GIS and DNAs. Thus, the reviewer doesn’t think the current cell experiment results support the conclusions.
The concentrations of pillar[5]arenes 1–4 (1 × 10–5 M) and GIS (3.47 × 10–2 M) used to assess cell viability actually differed by several orders. However, the presented approach in the manuscript opens up a possibility of creating pillar[5]arene-based herbicide antidotes.
In this regard, the sentence “It confirms the possibility of using these macrocycles as a glyphosate antidote” was replaced by “The developed approach opens up the possibility of creating pillar[5]arene-based herbicide antidotes” in the main text of the article and in the conclusion. All changes are highlighted in green.
- In table 3, the IC50 of GIS is much higher than pillararenes. Why do the authors use more toxic compounds to inhibit another binding and avoid DNA damage if they have similar binding affinity to the DNA?
Indeed, the IC50 of glyphosate is several orders higher than that of pillar[5]arenes. The presented work is a proof of concept, i.e. it is an implementation of an idea, not a finished product. The aim of the work was to show an inhibition of DNA interaction by glyphosate. It is necessary to select a structure of macrocycles in such a way as to reduce their toxicity for a subsequent development of antidotes based on pillar[5]arenes. After that, it will be possible to talk about real candidates for antidotes and about real objects. This manuscript represents an idea, which worked in the case of one of these pillar[5]arenes.
- In Figure 7, how about the control experiment data from DNA only?
It is well known that DNA dilution leads to an increase in the UV spectrum of the biomolecule. The concentration and the volume of the (macrocycle–DNA) associate remained constant (CPA = 3 × 10–6 M, CDNA = 2.5 × 10–7 M), while the herbicide concentration increased from the ratio of 1:0.3 to 1:10 relative to the concentration of the associate in the course of spectrophotometric titration for all systems. The solution volume remained constant (V = 3mL) throughout the experiment. The association spectrum (DNA-pillararene) did not change during the day.
The necessary changes have been made in the manuscript and have been highlighted in green.

Reviewer 4 Report
The manuscript presents the results regarding the synthesis for the first time of water-soluble betaine derivatives of pillar[5]arene containing amino acid residues (GlyGly and L-Phe) and possible application of these derivatives for nucleic acids protection against the damage by glyphosate isopropylamine salt (GIS).
The manuscript is well written, presents very interesting results in the field of the problem of herbicide toxicity.
Please use lgKa or lgKass throughout the manuscript.
In my opinion, the number of figures in Supplementary material is too high. You can give only one example with BindFit data. If for journal is ok, you can leave all figures.
I recommend the publication of this paper in IJMS journal.
Author Response
The manuscript presents the results regarding the synthesis for the first time of water-soluble betaine derivatives of pillar[5]arene containing amino acid residues (GlyGly and L-Phe) and possible application of these derivatives for nucleic acids protection against the damage by glyphosate isopropylamine salt (GIS).
The manuscript is well written, presents very interesting results in the field of the problem of herbicide toxicity.
First of all, we would like to thank respected Reviewer for careful consideration of the manuscript. In accordance with the comments, the following changes have been made:
- Please use lgKa or lgKass throughout the manuscript.
lgKass has been changed to lgKa throughout the manuscript and have been highlighted in green.
- In my opinion, the number of figures in Supplementary material is too high. You can give only one example with BindFit data. If for journal is ok, you can leave all figures.
The authors are absolutely convinced of the necessity of these Bindfit data, which are needed to verify the presented data in the manuscript.